# Culture and COVID-19: Impact of Cross-Cultural Dimensions on Behavioral Responses

**Nisha Nair [1,*,†]**, **Patturaja Selvaraj [2,†]** and **Ranjeet Nambudiri [2,3,†]**

1  Katz Graduate School of Business, University of Pittsburgh, Pittsburgh, PA 15260, USA
2  Management Department, Gettysburg College, Gettysburg, PA 17325, USA; pselvara@gettysburg.edu (P.S.); ranjeet@iimidr.ac.in (R.N.)
3  Indian Institute of Management, Indore 453556, India
*  Correspondence: nnair@pitt.edu
†  All authors are joint first authors.

**Definition:** The global pandemic of COVID-19 has impacted every sphere of human life across all nations of the world. Countries adapted and responded to the crisis in different ways with varied outcomes and different degrees of success in mitigation efforts. Studies have examined institutional and policy-based responses to the pandemic. However, to gain a holistic understanding of the pandemic response strategy and its effectiveness, it is also important to understand the cultural foundations of a society driving its response behavior. Towards that end, this entry focuses on a few key cultural dimensions of difference across countries and proposes that national culture is related to the protective behavior adopted by societies during COVID-19. The cultural dimensions examined in relation to COVID-19 include the dimensions of individualism vs. collectivism, power distance, uncertainty avoidance, masculinity and femininity, and future orientation. Inferences are drawn from academic research, published data, and discernible indicators of social behavior. The entry provides pointers for each dimension of culture and proposes that cultural awareness be made an important element of policy making while responding to crises such as COVID-19.

**Keywords:** culture; COVID-19; pandemic; cultural dimensions; individualism vs. collectivism; power distance; uncertainty avoidance; masculinity and femininity; future orientation

## 1. Introduction

For many, the unprecedented COVID-19 pandemic, which began spreading across the globe in January 2020, has been the first and only pandemic people have witnessed in their lifetime [1]. The pandemic has been a crisis of unprecedented proportion which brought with it conditions never encountered by the current generations. Governments and people across the world have been left scrambling to contain the spread of the virus and to adopt effective risk mitigation strategies, even after pharmaceutical interventions such as vaccines have come on the horizon [2]. The scale of damage that the SARS-CoV-2 virus has unleashed in terms of the lives lost, harmful mental and physical health consequences [3], and constraints on public health systems [4] has been unprecedented. As of May 2022, more than 520 million cases of COVID-19 and 6.2 million fatalities have been reported worldwide, with the United States, India, Brazil, France, Germany, the United Kingdom, and Russia having reported the largest number of cases [5]. On the other hand, countries such as New Zealand, Egypt, Sri Lanka, and Norway reportedly had fewer than 1.2 million cases [5]. Governments have exhibited wide variations in their responses to the pandemic. Some countries experienced nation-wide lockdowns such as India and China, while others adopted softer approaches such as that of Sweden [6–9]. Similarly, the societal response to the pandemic has also varied significantly. It can be argued that to understand a COVID-19 response strategy it is important to understand the interaction of both the formal

(institutional mechanisms) and the informal (cultural underpinnings) elements guiding decision making [10].

There is increasing recognition today that one cannot understand pandemic responses without adopting a behavioral science approach [11–13]. Consequently, there is a mounting body of work in this direction with several published works focusing on understanding responses to the COVID-19 pandemic through a behavioral lens [13,14]. Early in the pandemic, when pharmaceutical interventions were limited, some researchers called for recognizing and changing behavior to control the transmission of the virus [12]. Several studies have emerged pointing to differences in country-level responses. For example, one such study [9] points to differences in cross-country perceptions of risk influencing social distancing amidst the pandemic.

Different cultures react and behave differently to threat perceptions based on shared belief systems [15]. Differences is how people behave in a social context are often grounded in varying expectations and learned behavior acquired through socialization [16]. National cultures shape people's behaviors [15,17,18] and have been shown to predict people's wellbeing amidst the pandemic [19]. This is particularly relevant in times of crisis such as the pandemic when people tend to adhere more strongly to prevailing social norms [20]. Indicative of the powerful effect of culture in shaping behavior amidst the pandemic, in a recent study using econometric data, culture has in fact been shown to act as a substitute for state action in ensuring compliance with COVID-19 policies [21]. Thus, if culture has the potency to supplement or replace policy, it merits attention to understand the cultural dimensions shaping and guiding behavior, which can augment policy-level interventions to manage the ongoing pandemic.

Culture has been widely studied for many decades. Geert Hofstede's cultural diversity model is considered one of the major frameworks for understanding culture [15–17]. Since human behavior is a reflection of the underlying values that people subscribe to [22], it is worthwhile to reflect on the values that shape this behavior at the cross-cultural level. One of the earliest ways of understanding cross-cultural values or dimensions of difference was offered by Hofstede based on data from across 64 countries [17]. Hofstede conceptualized cultural differences emerging from differences in values categorized along certain dimensions [16,23]. The original four dimensions of difference across cultures were identified as individualism versus collectivism, power distance, uncertainty avoidance, and masculinity versus femininity [16,23]. Other dimensions such as long-term versus short-term were later included in the conceptualization. Individualism versus collectivism captures the dichotomy of independence versus interdependence, or loyalty to oneself compared to that towards the group [24,25]. It refers to the extent to which people affiliate with loosely or tightly knit social groups [25]. Power distance refers to the acceptance of power differentials in a society, while uncertainty avoidance refers to the extent of perceived discomfort with ambiguities or uncertainties in a society [16,18,26]. Masculinity versus femininity refers to the extent to which cultures prefer equality or egalitarianism between the sexes in a society. It captures the degree to which cultures are prone towards competition and assertiveness, or caring and nurturance [16,26]. The long-term versus short-term dimension refers to a cultural orientation that is rooted in either the present or the future [27].

This entry explores some of the above dimensions of cross-cultural difference as it relates to understanding or shaping behavior amidst the pandemic. In the following sections, the entry examines the relevant cultural dimensions and attempt to synthesize emerging literature linking cultural dimensions to the pandemic. The effect of cultural dimensions on variations in behavior will also be illustrated through some country-level differences, where available. In offering this overview of culture and COVID-19, we aim to understand the relevant cultural variables influencing and guiding human behavior at the collective level amidst a crisis such as the current pandemic.

## 2. COVID-19 and Cultural Differences between Individualism and Collectivism

A wealth of cross-cultural research has established the individualism versus collectivism dimension as one of the most vital dimensions capable of explaining differences between countries [28]. Being the most used dimension in cross-cultural research, individualism/collectivism is also thought to have the greatest predictive power [29].

The measure of individualism/collectivism is an indication of whether people tend to value individual goals, interests, and needs, or emphasize collective concerns and goals [24,25]. In other words, individualistic cultures focus on prioritizing personal needs above that of the group [16] while collectivist cultures tend to put more emphasis on group harmony over personal needs and interests [30]. As a cultural value, collectivism relates to ties between people. People who subscribe to a collectivist social value tend to interact cooperatively, embodying interdependence and recognizing mutual duties and obligations to one another [25]. Thus, collectivism is characterized by a strong sense of community, loyalty, trust, and respect towards one another and willingness to forgo individual needs in the interest of collective needs and goals. Individualistic societies on the other hand tend to value personal freedom and do not feel a compulsion to adhere to collective norms or goals.

Prior research has shown that collectivists are more likely to display adaptive responses in times of crisis [31]. Research has also established that in collectivist cultures people tend to adhere to social norms and demonstrate greater compliance as opposed to individualistic cultures [32]. A collectivist belief orientation would urge the protection of collective interests over efforts to maintain independent selves [33]. Further, given the perceived threat of community-based sanctions [34] for violating social norms in collectivist societies, people are more likely to participate in mitigation efforts during the pandemic to limit the growth of COVID-19.

Emerging studies are supporting the view that a collectivistic mindset drives willingness to engage in behaviors that can curtail the spread of the virus. Cultural factors were found to account for a large portion of the explanatory power for variation in COVID-19 containment efforts across nations [14]. Research has shown that individualists and collectivists tend to behave differently as far as virus containment measures such as social distancing is concerned [20]. Empirical studies are emerging showing behaviors such as mask wearing, social distancing, and intention to vaccinate are connected to collectivist mindsets or collectivistic cultures [20,35,36]. Using data from 67 countries varying on individualism and collectivism, Lu and colleagues [36] have been able to show that mask wearing was more common in countries high in collectivism and less so where individualism was high. Whether variations in individualism–collectivism led to different outcomes in terms of case rate and mortality rate was investigated in another research [14] across 54 countries, indicating that collectivism had a significant impact on containing the pandemic and on flattening the curve.

Individualism, on the other hand, has been linked to negative outcomes in relation to the COVID-19 pandemic. Examining intentions to reduce the spread of the virus, a study [20] found that individualism negatively predicted intention to engage in social distancing, mediated through belief in conspiracy theories and powerlessness. Early in the pandemic, cultures that were high on individualism were found to experience higher infected cases and death than collectivist cultures [14]. Other studies have shown that individualist cultures have contributed to worsening fatality rates, compared to nations with a collectivist culture [37]. The cultural norm of individualism has also been shown to have a positive effect on the growth rate of COVID-19 cases based on a time series analysis of data from 107 countries [8]. More recently, individualism has been linked to higher disease prevalence [21] and a higher mortality rate in a study across 110 countries considering climate risk and the COVID-19 mortality rate [38]. Another study [6] using data from 111 countries examined variations in behavior in relation to lockdown measures instituted by governments to curb the spread of the virus, finding support for the view

that individualism has made government interventions harder, as opposed to a collectivist dynamic which is more geared towards promoting compliance with COVID-19 policies.

Even at the level of individual experience of the pandemic's effects, individualism has been linked to higher experienced stress and isolation [19]. Some emerging research on the experience of social isolation amidst an ongoing pandemic suggests that collectivist societies are better equipped to deal with social isolation beget by pandemic conditions, with worry, fear, and anxiety being kept in check through networks of social connections [3]. Recent research has in fact shown that an individualist mindset led to a reduced intention to vaccinate compared with collectivism [35].

Looking at country-level data, the United States has recorded the largest number of infected cases and suffered a disproportionately higher number of lost lives to COVID-19 compared to all other countries in the first year of the pandemic [39]. Americans typically tend to be high on individualism [17,25,40], with American society marked by a preference for independence and a prioritization of individual freedom over collective welfare. In fact, as some note, there is even a resistance against the collective [41]. It can be argued that the fatalities suffered in the United States owing to the COVID-19 pandemic are partly attributable to the premium placed on individual freedom, independence, and autonomy, all captured through the cultural orientation of individualism [39]. This focus on individualism may have prompted a lack of adherence to public health measures in pursuit of individual freedom and liberty. People in individualistic societies tend to prioritize their own selves and wellbeing over those of others. There were even news reports [42] in the early days of the pandemic of beaches in Florida being crammed with people enjoying their spring break, with little thought to health advisories such as social distancing to mitigate the virus spread. In the US, even well-established practices such as mask wearing for containing the pandemic [43] have been met with resistance and have been portrayed as an affront to people's freedoms [39]. Thus, with the prevalent individualist ethos, what we may have seen in the US is the prioritization of individual choice and personal freedom at the cost of collective consequences.

On the other hand, collectivist countries such as South Korea and Singapore emphasized civic efforts such as persistent mask wearing [44], while individualistic nations such as the US and Italy found it challenging to encourage the adoption of mitigation efforts from early on in the pandemic [45]. For another collectivist country, Slovakia, some research [46] has shown a link between collectivism among Slovaks and containing the pandemic through engagement in prosocial behavior. Similarly, in collectivist China, there appeared to be greater adherence to lockdown rules [6] compared to the United States with its high individualism and lesser willingness to abide by COVID-19 policy recommendations. Considering a month-long period of government intervention on the case rate and pandemic-induced mortality rate, researchers [14] found that collectivist countries such as China, Japan, South Korea, and Singapore witnessed fewer cases compared to the more individualistic Western European and North American cultures. Based on emerging research, it can therefore be inferred:

*Key Takeaway: The cultural dimension of collectivism is positively related to higher engagement in COVID-19 protective behaviors such as mask wearing and social contact avoidance, compared to more individualistic cultures.*

Behaviors aligned with individualism are more likely to be harmful during times of crises with intensifying cases in individualistic countries emerging from an unwillingness to sacrifice individual needs or give up personal freedoms in the interest of collective welfare. By contrast, in collectivist societies, people tend to act as a group, with group success and objectives taking precedence over individual liberties or autonomy. There appears to be higher altruism and more effective rule observation in a collectivistic dynamic, which makes the implementation of COVID-19 policies such as lockdowns or mask wearing easier and more uniform in their adoption across a society. Highlighting collectivism and concern for others by focusing on the individual's role in benefiting the larger good can thus be a powerful narrative in shaping people's sensibilities to contribute towards efforts in limiting

the spread of the virus. It can encourage mitigation efforts and the willingness of a society to collectively curb the growth of the pandemic.

### 3. COVID-19 and the Cultural Norm of Power Distance

One of the dimensions of cross-cultural difference is power distance, which is an indication of people's attitude to power. Hofstede [26] argues that some countries are more unequal than others, with a high power distance culture characterized by higher deference shown to authority and authority figures, compared with those in low power distance cultures. It is indicative of the extent to which individuals with less power in society accept and expect that power is distributed unequally [47] and how much they endorse authority and power differences in a society [48]. Thus, power distance refers to how people react and behave in relation to power and power differentials in a society.

Power distance has been positively linked to conformity [49]. Individuals in high power distance countries are more willing to accept and observe rules from authority figures [48] and tend to be more willing to follow guidelines [50]. People in low power distance countries tend to be more willing to challenge authority and are generally perceived as being less submissive than individuals in high power distance cultures [51]. Countries higher in power distance are thus able to exercise greater authoritarian behavior and elicit desired compliance, compared to countries lower in power distance.

There is some emerging evidence [52] that in societies with high power distance, people's responses to mobility restrictions instituted by governments amidst the COVID-19 pandemic have generally been more favorable, compared with countries having a lower power distance. Subscribing to the values of power distance means that individuals are willing to acknowledge the role of the government and willingly comply with many tough measures instituted amidst the pandemic such as protracted lockdowns [7]. In another recent study [53], the moderating role of power distance was studied among consumers in relation to compliance with expected behaviors during the pandemic such as mask wearing. Results indicate that when countries have high power distance, along with other intervening factors, it can lead to greater compliance with expected behaviors. Since high power distance countries are more likely to evidence obedience and willingness to be led by others [54], there will likely be more effective participation in risk mitigation efforts, while countries low in power distance, being characterized by an emphasis on free will and with a greater predilection to challenge experts [55], are less likely to be effective in mitigation efforts.

In high power distant and collectivist countries such as South Korea, China, India, Singapore, Taiwan, the Philippines, and Vietnam, the high power distance coupled with high collectivism can create conditions for better enforceability of government-run initiatives to control the spread of the virus, as some researchers have proposed [7,56]. In both Vietnam and South Korea, surveillance apps were used by the government amidst the pandemic [44,57] and has been met with little resistance. In countries such as South Korea, China, India, Singapore, Taiwan, and Vietnam, people have been willing to comply with government-led measures such as downloading apps on their mobile phones, granting health authorities access to detail call records for contact tracing, and other such surveillance of the populace, geared towards curbing the spread of the virus [7]. It is likely to be the high power distance which has allowed countries such as Taiwan to institute something like an 'electronic fence' to track an individual's location via GPS and mobile phones, inviting police intervention whenever there is non-compliance to government measures put in place to curb the spread of COVID-19 [7]. There also appears to be high approval among the public for such practices in Taiwan even with hefty fines as high as $33,000 for non-compliance, given the high power distance and cultural beliefs that such sacrifices are necessary to keep the country safe [58].

In contrast, countries low in power distance such as the US, Italy, Norway, and Germany, where governments are viewed less deferentially, have had limited success in using governmental measures to curb the virus spread. Even with a technologically

advanced app for contact tracing such as the one developed in Germany, the app could not be mandated but only voluntarily used [59]. With its low power distance, both the US and Germany saw people protesting mitigation efforts [56]. In Norway too such an app for contact tracing had to be discontinued after its initial launch in the face of public disapproval and criticism [60]. Given the extremely low power distance in the US and strong data protection and privacy laws preventing government use of individual data [61], no such app even took shape in the US. In a separate study [52], it was demonstrated that government-induced social contact restrictions were higher in India and Namibia (high power distant countries) compared to the US or Germany (countries with low power distance). Another study of several European countries showed that power distance had a significant negative effect on the rate of increase in COVID-19 [62]. In separate research [8], it was observed that government intervention weakened the growth rate of virus transmission, and this effect was most significant in collectivist and high power distant countries, therefore suggesting that high power distance can have a moderating effect on the efficacy of other emergency response measures adopted by governments in times of crises, such as the COVID-19 pandemic. Thus, we can conclude:

*Key Takeaway: Adherence to health policy guidelines and government mandates and restrictions to curb the spread of COVID-19 is higher in high power distance countries than in low power distance countries.*

As discussed above, differences in behavioral responses in terms of abiding by preventive measures during the pandemic can be attributed to power distance norms prevalent in a country. High power distance ensures better enforceability of government diktats such as lockdowns, mask wearing, or quarantine rules put in place to control the spread of the virus. When this is combined with high collectivism, there tends to be high support for government mandates and less overall resistance to tough measures such as surveillance or extended lockdowns, as the welfare of the group takes precedence over individual rights and autonomy. In high power distance countries, even seemingly harsh measures invoked during the pandemic such as surveillance of the citizenry are rationalized by the populace as a necessary tradeoffs of relinquishing individual liberties to ensure the safety and wellbeing of the collective. Even with the availability of pharmaceutical interventions such as vaccination, in low power distant countries where there is a greater tendency to question or challenge authority [51], adoption rates can suffer as has been evidenced in countries such as the United States [2]. High power distance in effect allows governments to be more authoritative and pursue more stringent measures. Nations high in power distance have an advantage in containing the pandemic, as they face lesser resistance from the public to policy measures that might infringe on people's freedoms, and people are generally more willing to comply with necessary restrictions and interventions required to contain the spread of the outbreak.

## 4. COVID-19 and the Cultural Dimension of Uncertainty Avoidance

Uncertainty avoidance deals with the tolerance that a society has for ambiguous conditions. It has been noted [17,26] that members of certain societies are more comfortable in predictable and structured situations (high uncertainty avoidance) while other societies seem to thrive in conditions of ambiguity and unpredictability (low uncertainty avoidance).

Several studies have examined the outcomes and correlates of uncertainty avoidance, indicating that high uncertainty avoidance is associated with a lower propensity for risk and a greater desire for expectedness. In a study with 303 participants from Spain, Germany, and Sweden [63], it was noted that customers who came from high uncertainty avoidance cultures were less satisfied with service defect situations than those from low uncertainty avoidance cultures. This points towards a greater tolerance for ambiguity in low uncertainty avoidance cultures. Societies that are high on uncertainty avoidance are characterized by higher stress and lower scores on subjective wellbeing [26]. In such societies, individuals have shown a lower propensity to switch jobs since new and different roles are often considered threats, with people attempting to bring a sense of stability and predictability

in their lives [64]. It has been posited [65] that uncertainty avoidance is an antecedent of risk perception, with decision makers in such cultures preferring the conservative choice in order to increase predictability and control [66]. Uncertainty avoidance has also been studied as a moderator of the relationship between leadership behavior and innovative work, suggesting a stronger correlation in high uncertainty avoidance cultures [67]. With samples drawn from India, the United States, Brazil and the United Kingdom, a separate study [68] found that uncertainty avoidance moderated the relationship between intention to use mobile banking and its actual use, with a stronger relationship found in higher uncertainty avoidance cultures. Another study [69] conducted with participants from a high uncertainty avoidance culture (Turkey) and a low uncertainty avoidance culture (Malaysia) found that the cultural dimension had an influence on customer loyalty. A study conducted in Norway (known to be moderate in uncertainty avoidance) showed that uncertainty avoidance was a strong predictor of interest in valid and established genetic testing methods for highly penetrative diseases [70]. This finding assumes significance for the current essay which attempts to understand the relationship between dimensions of national culture and the COVID response behavior of societies.

Unexpected events with a global impact (such as the pandemic, a financial market meltdown, an oil crisis, etc.) create conditions of ambiguity and carry a high risk of decision failures. Given that uncertainty avoidance cultures are prone to risk aversion and conservatism in decision making [64,65], it seems reasonable to argue that the uncertainty avoidance dimension of a national culture is related to the COVID-19 protective behaviors adopted by nations and societies. Erman and Medeiros [71] gathered COVID-19-related data from over 73 countries and adopted meta-analytic methods to show that uncertainty avoidance and other cultural dimensions were independent predictors of test positivity, case infection, and mortality risk. Separately, a linkage between uncertainty avoidance and higher rates of COVID-19 infection has been suggested [64] with the argument that countries high in uncertainty avoidance are more likely to resist changes in social structure. This would render strategies such as mandatory mask wearing and social contact avoidance less efficient, thus increasing infection rates from the virus. Uncertainty avoidance has also been linked to the risk perception of tourists in the COVID-19 context [72,73]. Another study posited uncertainty avoidance to moderate the relationship between the stringency of lockdown processes and country-level innovation, albeit with limited support for the relationship [74].

Considering uncertainty avoidance scores for a few countries [75], it appears that uncertainty avoidance may have a correlation with the COVID-19 protective behaviors of societies. A recent study by the University of Michigan [76] classified Vietnam, a culture low in uncertainty avoidance, as among the few nations which exhibited a rapid public health response by successfully imposing social and behavioral changes such as social contact avoidance and mandatory mask wearing. Similarly, Singapore, considered low in uncertainty avoidance [75], saw the government imposing high monitoring and control measures [77]. For instance, students who reported back to the colleges were asked to record and report their temperature several times in a day and entry into public places such as the libraries and cafeterias were strictly regulated for a longer duration [77]. Aggressive contact tracing measures such as scanning the identification documents of people at supermarkets and other public locations were part of a centralized control approach [77]. This seems in line with the view that low uncertainty avoidance cultures are more likely to be open to stricter control measures. On the other hand, Germany, a high uncertainty avoidance society [77], witnessed many anti-lockdown protests during the pandemic, indicating that high uncertainty avoidance cultures are less likely to impose stringent measures such as lockdowns or mandatory mask wearing. The following takeaway therefore emerges:

*Key Takeaway: Cultures low in uncertainty avoidance are more likely to have a tolerance for varied risk mitigation strategies requiring behavioral changes than cultures high in uncertainty avoidance.*

In the initial stages of the pandemic when vaccines were still under development, the only method of preventing the spread of the virus was through behavioral changes in society. This included measures such as social contact avoidance, mandatory mask wearing, repeated washing of the hands, and avoiding public areas. Government interventions included lockdowns of varying intensity and duration as well as strict regulation of the above-mentioned measures. It can be argued that acceptance or resistance of social and behavioral changes induced by the pandemic are likely to be affected by the uncertainty avoidance propensities of societies. Thus, nations low on uncertainty avoidance found it simpler to implement stringent social and behavioral changes during the pandemic. People in low uncertainty avoidance cultures tend to tolerate uncertainty and ambiguities better than those in high uncertainty avoidance cultures. Hence, it becomes easier to introduce varied policy interventions without backlash or resistance from people.

## 5. COVID-19 and Cultural Orientations of Masculinity and Femininity

Masculinity and femininity denote differences in cultures based on perceived masculine and feminine values. According to Hofstede [47], masculinity and femininity are in relation to differences in emotional roles between the sexes. The values of assertiveness and competition are categorized as masculine, and the values of care, compassion, and modesty as feminine in this conceptualization [26,47]. "Cultural masculinity stands for a focus on ego, money, things, and work; cultural femininity for a focus on relationships, people, and quality of life." [47] (p. 72). It is argued that "women in feminine countries have the same modest, caring values as the men; in masculine countries, they are somewhat assertive and competitive, but not as much as the men, so that these countries show a gap between men's values and women's values." [47] (p. 63). Thus, a culture high in masculinity is interpreted as one where there are differences in values and work roles along gendered lines, while cultures high in femininity can be considered as more egalitarian and less polarized in values between the genders. Countries high on masculinity tend to focus on material, career, and business success, with cooperative behavior not much appreciated in such societies, while countries high on femininity tend to focus on modesty and give importance to quality of life, kindness, and empathy [78]. Feminine societies are thus thought to stress the importance of a work–life balance, have minimal emotional and social role differentiation between the genders, express sympathy and support for the weaker in society, and have the values of caring and being modest subscribed to by both genders [26], whereas masculine societies support explicit emotional and social differentiation amongst the genders, have an admiration for stronger people, and have work deemed as central, to an extent where it often takes precedence over the family [26].

Research has shown masculinity to be high in Japan, Germany, Austria, and Switzerland, and as moderately high in English-speaking Western countries such as the United Kingdom and the United States [26]. For example, South Koreans are considered to be less assertive and competitive than Americans, thus scoring lesser in masculinity than the United States [79]. Femininity is deemed high in the Netherlands and Nordic countries such as Norway, Sweden, and Denmark, and moderately high in countries such as France, Chile, and Thailand [26,47].

The United States (high on masculinity) and Norway and Denmark (high on femininity) handled COVID-19 very differently. Norway acted fast and went into lockdown in mid-March of 2020. It closed schools, restaurants, gyms, and other public places and banned cultural and sports activities and the entry of people from other countries [80]. It resulted in a lower number of people being infected and in a lower death rate per capita in comparison to the United States [80]. In the US, the federal government did not mandate the closure of schools unilaterally but left it to the individual schools to assess the situation and decide about the closure [81]. When some states in the US started introducing masking requirements, there was resistance to follow the mask mandates at the individual level [82]. Additionally, in the US, the federal government and state governments were sending conflicting and mixed messages [83]. Overall, resistance to the pandemic-induced

restriction was high in the US. In contrast, the robust social security system in Norway with free-cost education, a public financed health care system, paid maternity and parental leave, and low-cost childcare (making it a more egalitarian society and high on feminine cultural norms) helped Norway tide over the pandemic with much more ease than many other countries [83,84]. In the United States, the then President faced major criticisms for his failure to protect the people of the U.S. from the COVID-19 pandemic [85], while the prime minister of Norway was lauded for handling the crisis with kindness and compassion [83,85,86]. Another Scandinavian country, Denmark, a feminine society with a smaller gap between rich and poor, was less hit by the pandemic [87]. Denmark registered fewer infected cases and deaths compared to the United States [87,88]. Being one of the first countries in Europe to take swift action to ban large gatherings and impose restrictions on travel, Denmark saw the closure of daycares, schools, and universities early on in the pandemic [87]. Given the preference for work–life balance, a feminine cultural norm [17], people in Denmark took to the new normal fairly quickly and smoothly, making the adoption of the pandemic-induced restrictions easier [87]. In contrast, in the United States, with the fear of job loss, expensive college education and health care systems, and rising fears of economic insecurity [89], there was greater resistance to pandemic-imposed restrictions [89]. This entry can therefore surmise the following:

*Key Takeaway: Feminine cultural norms enable higher receptivity and societal tolerance for pandemic-induced shocks compared to masculine cultural norms.*

Feminine dominant societies prioritize concern for the weaker community members and also elevate women to top managerial and political positions [86]. For countries like Norway and Denmark, with a dominant feminine culture, there also tends to be wider societal support for policies geared towards the larger public good [86]. Pandemic-induced restrictions such as forced work from home also allowed for a better work–family integration, a value cherished in societies that follow feminine cultural norms.

## 6. COVID-19 and the Cultural Dimension of Long-Term versus Short-Term Focus

One of the ways in which cultures differ is grounded in how people perceive time. Differences in time orientation has thus emerged as another dimension of cross-cultural difference [90]. One set of this difference refers to the distinction between monochronic and polychronic cultures, and another captures a differing focus on the past, present, and future [90,91]. A past orientation emphasizes a view towards the past and focuses on the maintenance or restoration of traditions, while a future orientation stresses a mindful awareness of the future and actions and behaviors grounded in things such as planning and delayed gratification, with a view towards ensuring a better future [92]. A future orientation is akin to Hofstede's conceptualization of the cultural dimension of long-term versus short-term focus [27].

Cultures high on short-term orientation show a preference for immediate rewards and individualism, while those with a longer-term orientation tend to focus on long-term goals and are more likely to be collectivist in orientation [27]. Having a future orientation has also been tied to protective behaviors such as engaging in social contact avoidance [52]. There is some evidence that a future orientation is linked to people's abilities to set goals for their future health [93]; thus, it should follow that even with regard to the pandemic, a future orientation should be instrumental in setting goals that move the needle towards a pandemic-free world. A case in point is China, which is high on long-term orientation [18,24]. The country was among the first to use technology aids such as facial recognition for identifying and isolating affected individuals [7] as well as implementing rapid nucleic acid testing, which seems to indicate a futuristic orientation towards prevention rather than containment. There is also some emerging research that suggests countries higher in long-term orientation have been better able to avoid higher fatality rates compared with those having a preference for a short-term orientation [37].

Lu and colleagues [36] examined the masking behavior of people from 29 countries. These included countries high on future orientation such as South Korea and Singapore as

well as countries with a short-term orientation such as Canada and the United States. The results indicated that people in Singapore and South Korea showed a greater propensity to wear masks during the pandemic. Research indicated that countries with a high long-term orientation were able to successfully implement COVID-19 protocols such as social contact avoidance and mask wearing [7]. For instance, in South Korea and Singapore, the governments found it easier to implement social contact avoidance and persistent masking [44] while mandatory mask wearing was met with significant resistance in the United States [43]. As mentioned earlier, Taiwan, a nation with a high long-term orientation, was able to implement an "electronic fence" for surveillance and contact tracing [7]. Therefore, it can be surmised:

*Key Takeaway: A future or long-term orientation will be positively associated with increased protective behaviors amidst the pandemic.*

Considering a pandemic-free world as the aspired future state, cultures with a dominant future orientation are more likely to engage in practices and behaviors that help drive towards this desired future, even if it discomfits the present. The degree to which cultures have a long-term or future orientation could therefore suggest whether people are willing to sacrifice personal liberties and autonomy in the present, such as refraining from socializing, in order to move towards the desirable future of a pandemic-free existence.

## 7. Conclusions

Much has happened since the COVID-19 pandemic was first detected. In the early stages, prior to the development of vaccines, most countries deployed social and behavioral strategies of varying nature, including nation-wide or localized lockdowns, social contact avoidance, and mask wearing, among others. Different nations implemented travel restrictions of varying intensity and duration [94]. Access to vaccines also varied across nations, with the US and the UK making vaccines available much earlier than many other nations [95]. The pandemic itself has witnessed multiple waves across 2020–2022, with new mutations appearing [96]. Amidst this evolving situation, attention to cultural factors can go a long way in both understanding what is fueling behavior responses and designing effective interventions and policy recommendations.

Examining the role of cross-cultural factors in driving pandemic responses, our analysis revealed that cultural dispositions can either aid or thwart effective responses during times of crisis, such as that of the pandemic. Hofstede's [15–17] model has also been previously applied to understand behavior during other crisis contexts such as the global financial crisis [97]. Cultural characteristics such as individualism can be a drawback in times of emergencies such as the pandemic, where in the absence of effective checks for furthering individual interests, it can create conditions that can potentially accelerate the spread of the pandemic. There is emerging evidence that a cultural preference for individualism makes it challenging to promote risk mitigation behaviors such as vaccination and mask wearing [36]. During a global crisis, when concern for others drives pandemic containment behaviors such as social contact avoidance, mask wearing, vaccination, and adherence to risk mitigation strategies, an overt emphasis on seeking and fulfilling one's individual needs for autonomy and disregard for how one's actions could contribute towards the spread of the virus can be catastrophic. On the other hand, a collectivistic orientation has been a useful cultural trait in limiting the spread of the virus. Promoting collectivism, where possible, may be a way to strengthen policy measures to contain the pandemic and increase people's engagement with efforts in this direction [20].

Similarly, having high power distance as a cultural norm can be useful to ensure compliance with and the adoption of government mandates and policy recommendations in times of crises. Whatever be the drawbacks of having high power distance, in times of crises when expert guidelines and adherence to government mandates are crucial for crisis resolution, high power distance can come in handy to ensure high compliance and less resistance to authority.

Another cultural norm that seems to be advantageous in a time of crisis is a feminine cultural norm, with values of compassion and egalitarianism fueling adequate responses supported by the larger community. Feminine cultural norms potentially indicate a higher tolerance for absorbing shocks during a crisis situation compared with masculine cultures.

It also appears that nations with high uncertainty avoidance faced challenges while implementing control measures for managing the pandemic such as lockdown, curfews, social contact avoidance, and mandatory mask wearing. Owing to the preference for order and stability, any initiatives or attempts to manage the crisis that is shrouded in further uncertainty is likely to be met with some degree of resistance in high uncertainty avoidance cultures. Thus, policy makers are advised to recognize that having a sound rationalization for instituting measures for risk mitigation may not be sufficient in itself if it does not address peoples' misgivings or concerns with ambiguity.

Finally, the cultural value of future orientation or long-term focus can also come in handy to make the many inconveniences necessitated for the management of the pandemic, such as restrictions on personal mobility, tolerable. Mobilizing individual action to engage in such sacrifices are also made easy when there is a long-term cultural orientation. A long-term cultural orientation can thus help build resilience in the face of an ongoing crisis, making tough decisions more palatable.

Thus, what can be inferred from exploring the cultural underpinnings of behavior is that certain cultural norms tend to be more conducive to the facilitation of risk mitigation strategies versus others, such as collectivist orientations, high power distance, low uncertainty avoidance, feminine cultural norms, and a long-term or future cultural orientation.

In arguing for considering the role of cultural differences in shaping pandemic behavior, while looking at the cultural differences driving divergent behavioral responses, the entry has illustrated with a few country-level responses wherever information is available and the linkage to behavior can be inferred. The country-level illustrations are only indicative and are in no way meant to be exhaustive or to be interpreted as a suggestion that all countries with similar cultural norms will behave in analogous ways. Thus, this entry does not purport to explain behavioral responses for all countries using the cultural model of Hofstede or make the claim that all countries subscribing to the same cultural norm will behave the same way. There will surely be instances when a particular country meets with less or more success in tackling the pandemic than other countries with similar cultural dispositions. Clearly, there are numerous other factors that influence effectiveness of pandemic responses such as economic factors, population density and size of a country, vaccination availability, governmental action/non action, and effective public health systems, among others. The larger point is that culture is one of the factors that appears to have some bearing on how groups of people behave amidst the pandemic and should be considered as another useful tool in a country's arsenal to fight the pandemic.

The continually evolving nature of the pandemic must also be acknowledged and borne in mind. What is true of a country's response today may change in the future as the pandemic changes, the political will and resources available to fight the pandemic changes, and newer interventions continue to be developed. Therefore, while thinking of a country's pandemic response, it is useful to keep in mind that this is also susceptible to variation and influence by a number of other intervening factors, given the ongoing nature of the pandemic. Thus, the current entry is limited to understanding pandemic behavior at specific moments in time, but this literature is continually being added to and country-level responses and the management of the pandemic is also ever morphing even as this entry is being written.

While the entry is a commentary on the possible relationship between the cultural dimensions proposed by Hofstede [16,17] and the pandemic response behaviors of societies, it does not claim to be a propositional inventory but instead arguments are based on an integration of the extant literature and demonstrable information on the actual responses of societies. Given that pandemic response strategies have evolved with growing information about the virus [96], the entry recommends a longitudinal time lagged analysis of data to

draw firm conclusions about the nature of the relationship between cultural dimensions and COVID response behavior.

An understanding of how cultural norms influence people's behavior can be a useful tool in shaping and implementing policy interventions to manage the ongoing pandemic or the future of such crises. Our cultural analysis of behavior linked to COVID-19 offers one lens for thinking about what can enable or disrupt even the best thought out mitigation efforts amidst an ongoing crisis. Given differences in cultural orientations and ensuing the behavioral tendencies of people, a one-size-fits-all approach to navigating and managing the pandemic may not always be effective. Governments thus need to incorporate cultural awareness into their policy decisions for emergency preparedness and instituting containment measures and interventions, as the COVID-19 pandemic has shown us. Even with regard to any future global crisis, our analysis reveals that cultural norms can play a powerful role in disposing people to react and behave in particular ways. Effective risk mitigation strategies are therefore ones which factor in cultural norms to either use to their advantage or to actively design policy interventions to counter cultural tendencies that can subvert effective crisis resolution.

**Author Contributions:** Conceptualization, N.N., P.S. and R.N.; methodology, N.N., P.S. and R.N.; formal analysis, N.N., P.S. and R.N.; writing—original draft preparation, N.N., P.S. and R.N.; writing—review and editing, N.N., P.S. and R.N. All authors have read and agreed to the published version of the manuscript.

**Funding:** This research received no external funding.

**Institutional Review Board Statement:** Not applicable.

**Informed Consent Statement:** Not applicable.

**Conflicts of Interest:** The authors declare no conflict of interest.

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
