# Peer review of "Culture and COVID-19: Impact of Cross-Cultural Dimensions on Behavioral Responses"

_encyclopedia, doi:10.3390/encyclopedia2030081_

Round 1

Reviewer 1 Report

The authors submit an interesting review study for evaluation. I have practically no reservations about the content. As for the main positives of the paper, they are a clear structure and relatively good readability. When it comes to negatives, there are more. The topic as such has been generic at least for the last year. In my opinion, universal recommendations cannot be made in it (if the authors want to avoid overgeneralization). It is also debatable how many (and how good) references would need to be synthesized for such work, as several dozen studies are published daily. So, overall, a meaningful study needs to better summarize the results. Perhaps I would consider adding a section describing limitations, as it is not possible to ignore the fact that the topic is evolving and describing a global phenomenon. The question is whether the length of the submission is no longer beyond the length of other "entries". I cannot judge this fact responsibly. In conclusion - the quality of the presentation is adequate, the scientific soundness as well (as long as the limitations are described), the significance of the content is fine even, it is a bit obsolete. I also leave it to the authors to summarize the key findings across the individual chapters in the overview tables, and visualize the overall findings through diagram / infographic. I hope my comments will help.

Author Response

Thank you for your comments and observations. As per your suggestion, we have now included a separate section on limitations where we make note of your suggestions. We have also modified the conclusion accordingly.

Thank you for your feedback and hope you find the revision to your satisfaction.

Reviewer 2 Report

Dear authors,

The first impression reading your paper, impression which did not change through the end, was the logic composition and the concision of the article. It is also a well-written paper.

I am not specialized in the sociology domain, but I must acknowledge the clarity of the content.

The Conclusion section is the weakest part of the paper. Resuming the findings, this section does not highlight the implications and ways to counteract possible other global events with negative impact on the world population and states, does not explain the value and novelty of the findings and also, does not outline authors’ intentions towards future researches on this issue, based on the present article.

Author Response

Thank you for your comments and observations. Accordingly, we have now reworked our conclusion to highlight the implications and better summarize the findings of our analysis. As suggested, we also speak to future research in our limitations section which has now been added as a separate section.

Thank you for your feedback and hope you find the revision to your satisfaction.

Reviewer 3 Report

To Authors,

The entry manuscript by Nair et al describes pointers for cultural characteristics and propose that cultural awareness is an instrumental element in decision making while handling a crisis such as global pandemic, Covid-19.

The manuscript is interesting and my concerns are outlined below.

a) The current entry follows a model proposed by Hofstede. How successful this model is in terms of another crisis?

b) Direct references to naming the Presidents and Prime Ministers should be avoided. Instead simply using President and PM would be more appropriate.

c) This entry focusses more on Western countries. Can this model be translated to Asian, Middle Eastern and African Continents? Since African and Middle Eastern countries were even more successful in containing the virus despite being more masculine.

d) Since US and Canada are geographically large, would it be wise to compare to with geographically smaller nations where it is easier to enforce policies?

e) Can this model be translated to North Korea?

f) Very little has been discussed about nations like India and China who have used different methods to contain the virus.

g) Since cultures vary throughout each countries, how can authors  provide arguments to all western countries?

h) Due to humungous cultural variances, how can cultural awareness in the Governments decision making help pandemic crisis?

i) Sweden never introduced lockdown however was still successful in containing the virus than other nordic countries like Norway. Can authors comment on this?

j) The entry can include more data from other countries from all continents to have a nice overview and then make a reasonable conclusion.

Author Response

Our Overall Response: Thank you for your feedback and observations. We have tried to address all comments in our revision. The changes appear in the following sections: a newly added section titled ‘Limitations’, changes made to the ‘Conclusion’ section, changes made under Section 5 (section on Masculinity and Feminity cultural norm), and changes made under Section 6 (for long term versus short term orientation). We have also removed any mention of names of Presidents/Prime ministers as suggested.

Point by point response to the comments of Reviewer 3

  1. a) The current entry follows a model proposed by Hofstede. How successful this model is in terms of another crisis?

Response: We have tried to address this in our revision. The focus of our paper is specific to understanding cultural factors driving divergent behavioral responses in relation to the COVID-19 pandemic. In our limitations and conclusion sections, we speak about the limits of consideration for interpreting the findings. Additionally, based on your comment, we also looked up further literature to see the applicability of Hofstede’s model in relation to other crises. Accordingly, we have incorporated a note on use of Hofstede’s model in understanding behavior in relation to the global financial crises. This change appears in the conclusion section.

  1. b) Direct references to naming the Presidents and Prime Ministers should be avoided. Instead simply using President and PM would be more appropriate.

Response: Thank you for the observation. We have now eliminated direct reference to naming President/Prime Ministers in the revision.

  1. c) This entry focusses more on Western countries. Can this model be translated to Asian, Middle Eastern and African Continents? Since African and Middle Eastern countries were even more successful in containing the virus despite being more masculine.

Response: Our paper focuses on both western and non-western countries. We have made changes in our revision to include reference to more countries and we also discuss in our limitations and conclusion section about the applicability of the findings.

  1. d) Since US and Canada are geographically large, would it be wise to compare to with geographically smaller nations where it is easier to enforce policies?

Response: Thank you for the comment. We have tried to address this in our newly introduced limitations section where we also clarify that the cultural dimensions are but one factor impacting behavioral responses. We also acknowledge the role of other factors such as geographical size, population of a country, available resources, etc., in driving divergent pandemic response.

  1. e) Can this model be translated to North Korea?

Response: Please see our limitations section where we have clarified the scope of our findings.

  1. f) Very little has been discussed about nations like India and China who have used different methods to contain the virus.

Response: The focus of our paper has not been to explicitly explain differences in specific country responses to the pandemic, but rather looking at the general pattern of cultural underpinnings driving variance in behavior at the aggregate level. We acknowledge in our limitations section that there can be a number of other factors explaining differences in country level approaches and outcomes to containing the pandemic. We have included though mention of both India and China in our revision under relevant sections while discussing the different cultural dimensions.

  1. g) Since cultures vary throughout each countries, how can authors provide arguments to all western countries?

Response: As we mention in our paper, there are differences in how countries behave and respond to the pandemic based on the cultural dimensions of difference explored in this paper. In analyzing the different dimensions, we have not looked at specific countries or gone by a western or non-western country approach. In fact, in the country illustrations under our discussion for the different cultural dimensions being examined, we include both western and non-western countries, wherever there is evidenced literature and where adequate linkages to the cultural dimensions can be made. Please also see our limitations section in our revision where we have tried to clarify the scope of our paper.

  1. h) Due to humungous cultural variances, how can cultural awareness in the Governments decision making help pandemic crisis?

Response: We have tried to address this in the conclusion and limitations section.

  1. i) Sweden never introduced lockdown however was still successful in containing the virus than other nordic countries like Norway. Can authors comment on this?

Response: We acknowledge that countries with similar cultural norms can still have differing outcomes with regard to the pandemic. While cultural determinants of behavior can be one of the factors driving pandemic response behavior, it is clearly not the only one and many other intervening factors have a role to play, as we acknowledge in our limitations section. Our attempt in this entry paper is to point to one of the lenses through which pandemic responses can be analyzed and understood. Please also see our limitations section where we speak of the considerations and limits of interpreting the analysis of the current paper. 

  1. j) The entry can include more data from other countries from all continents to have a nice overview and then make a reasonable conclusion.

Response: Thank you for your comments. We have now revised the draft accordingly and refined our conclusion as well.

Thank you for all your comments and feedback. We hope you find the revision to your satisfaction.

Round 2

Reviewer 1 Report

I would recommend considering moving the chapter describing limitations to the chapter presenting the conclusion (at the end), or at the position of the last chapter of the study. I have no serious reservations about the rest of the text. The comments were incorporated to an acceptable extent.

Author Response

Thank you. We have now moved our limitations section to be part of our ‘Conclusions’ section and incorporated it at the end as suggested. Thank you for your feedback and hope you find the revision to your satisfaction.

Reviewer 2 Report

The article meets the requirements for publishing in its current form

Author Response

Thank you very much for your positive feedback and comments.

Reviewer 3 Report

The authors tried to address all the concerns and I have no further comments.

Author Response

(The authors gave the same response as above.)

Round 3

Reviewer 1 Report

The comments have been incorporated to a sufficient extent.